# Repeated Administration of Cisplatin Transforms Kidney Fibroblasts through G2/M Arrest and Cellular Senescence

**DOI:** 10.3390/cells11213472

**Published:** 2022-11-02

**Authors:** Jia-Bin Yu, Dong-Sun Lee, Babu J. Padanilam, Jinu Kim

**Affiliations:** 1Interdisciplinary Graduate Program in Advanced Convergence Technology & Science, Jeju National University, Jeju 63243, Korea; 2Subtropical/Tropical Organism Gene Bank, Jeju National University, Jeju 63243, Korea; 3Jeju Microbiome Research Center, Jeju National University, Jeju 63243, Korea; 4Faculty of Biotechnology, College of Applied Life Sciences, SARI, Jeju National University, Jeju 63243, Korea; 5Department of Urology, Tisch Cancer Institute, Icahn School of Medicine at Mount Sinai, New York, NY 10029, USA; 6Department of Anatomy, Jeju National University College of Medicine, Jeju 63243, Korea

**Keywords:** cisplatin, kidney fibroblast, myofibroblast, senescence, cell cycle, p21, lamin B1

## Abstract

Cisplatin is a potent chemotherapeutic used for the treatment of many types of cancer, but it has nephrotoxic side effects leading to acute kidney injury and subsequently chronic kidney disease (CKD). Previous work has focused on acute kidney tubular injury induced by cisplatin, whereas the chronic sequelae post-injury has not been well-explored. In the present study, we established a kidney fibroblast model of CKD induced by repeated administration of cisplatin (RAC) as a clinically relevant model. In NRK-49F rat kidney fibroblasts, RAC upregulated α-smooth muscle actin (α-SMA) and fibronectin proteins, suggesting that RAC induces kidney fibroblast-to-myofibroblast transformation. RAC also enhanced cell size, including the cell attachment surface area, nuclear area, and cell volume. Furthermore, RAC induced p21 expression and senescence-associated β-galactosidase activity, suggesting that kidney fibroblasts exposed to RAC develop a senescent phenotype. Inhibition of p21 reduced cellular senescence, hypertrophy, and myofibroblast transformation induced by RAC. Intriguingly, after RAC, kidney fibroblasts were arrested at the G2/M phase. Repeated treatment with paclitaxel as an inducer of G2/M arrest upregulated p21, α-SMA, and fibronectin in the kidney fibroblasts. Taken together, these data suggest that RAC transforms kidney fibroblasts into myofibroblasts through G2/M arrest and cellular senescence.

## 1. Introduction

Cisplatin is a chemotherapy medication commonly used to treat many solid cancers including metastatic testicular tumor, metastatic ovarian tumor, and advanced bladder cancer [1,2]. The mechanism of the anticancer action of cisplatin is a cancer-cell-specific formation of interstrand and intrastrand cross-links through binding between cisplatin and DNA, resulting in defective DNA templates, arrest of DNA synthesis and replication, and finally DNA damage [1,3]. Among many cancer treatment drugs approved by the United States Food and Drug Administration (FDA), cisplatin is the most compelling one, but approximately 30% of patients treated with a single dose of cisplatin develop kidney dysfunction, resulting in acute kidney injury (AKI) [4,5]. Because of that, cisplatin is repeatedly administered at low doses to reduce nephrotoxic side effects, but AKI eventually occurs [6]. Even more problematically, cisplatin-induced AKI can lead to chronic kidney disease (CKD), as maladaptive repair after acute injury to the kidney causes progression to CKD [7]. A few animal studies have unveiled that repeated administration of cisplatin (RAC) at a low dose (7 to 9 mg/kg body weight of mice) induces tubulointerstitial fibrosis, a hallmark of CKD [8,9,10]. A recent clinical study examining the long-term outcome after RAC has shown that most cancer patients have experienced a permanent decrement in kidney function after AKI [11]. However, the underlying mechanism facilitating progression from AKI to CKD after RAC remains largely unclear.

Kidney fibroblasts reside in the interstitial area between capillaries and tubules, and maintain the three-dimensional architecture of normal kidney tissues [12]. In the interstitial where the reabsorption and secretion of fluid and solutes happen across the tubulointerstitial compartment, kidney fibroblasts play a role in the production of extracellular adenosine [13] and erythropoietin [14]. During tubulointerstitial fibrogenesis, myofibroblasts are a primary source of an excessively deposited extracellular matrix, causing destruction of normal kidney tissue architecture and loss of kidney function [15]. The hallmark of myofibroblasts is expression of α-smooth muscle actin (α-SMA), which builds bundles of microfilaments with dense bodies [12]. The percentage of myofibroblasts is proportionally correlated with the severity of tubulointerstitial fibrosis [16]. Although circulating fibrocytes [17] and local cells, including tubular epithelial cells [18], endothelial cells [19], and macrophages [20], can transdifferentiate into α-SMA-positive myofibroblasts, resident kidney fibroblasts and microvascular pericytes are the primary source of myofibroblasts during tubulointerstitial fibrosis [21]. Thus, kidney fibroblasts play a crucial role in dictating tubular cell fate and the outcome of CKD. However, most of the previous studies on the mechanism of RAC-induced tubulointerstitial fibrosis focused on the injury to the parenchyma and kidney tubules in cisplatin nephrotoxicity (four RAC with 0, 0.5, 1, 2, or 5 μM in mouse proximal tubule epithelial cell line [9,22,23]. In this study, we hypothesized that RAC transforms resident kidney fibroblasts into myofibroblasts and induces fibrosis. To investigate this hypothesis, we established an in vitro model of RAC in resident kidney fibroblast cells and assessed myofibroblast characteristics including cellular senescence, hypertrophy, and cell cycle arrest. To further confirm our finding on the molecular mechanism of transformation of resident kidney fibroblasts into myofibroblasts, we tested pharmacological inhibitors of specific molecules in the established model. Our results verify that resident kidney fibroblasts are an important source of tubulointerstitial fibrosis and CKD induced by RAC.

## 2. Materials and Methods

### 2.1. Cell Culture and Treatment

The normal rat kidney fibroblast cell line (NRK-49F; American Type Culture Collection, Rockville, MD, USA; product no. CRL-1570) was cultured in Dulbecco’s Modified Eagle Medium high glucose (DMEM; Welgene, Gyeongsan, Gyeongsangbuk, Korea; product no. LM 001-05) supplemented with 10% fetal bovine serum (Welgene; product no. S 101-07), 100 units/mL penicillin, and 0.1 mg/mL streptomycin (Welgene; product no. LS 202-02) at 37 °C in 95% air and 5% CO_2_ incubator (Esco Technologies, Horsham, PA, USA; product no. CLM-170A-8), as described previously [24]. The cells were subcultured by 0.05% trypsin-EDTA (Welgene; product no. LS 015-01) when 80% confluence was reached and used after at least three passages. Sixteen to eighteen hours after subculture, the cells were starved for 2 h and subsequently treated with 0 to 30 μM *cis*-Diammineplatinum(II) dichloride (cisplatin; Sigma-Aldrich, St. Louis, MO, USA; product no. 479306) in normal culture media, 0 to 3 μM UC2288 (a cell-permeable p21 inhibitor; Sigma-Aldrich; product no. 5328130001 [25]) in 0.1% dimethyl sulfoxide (DMSO; Biosesang, Seongnam, Gyeonggi, Korea; product no. DR1022-500-00), or 0 to 10 nM paclitaxel (a microtubule-interfering agent [26]) in normal culture media. We treated them for 6 h, changed the treated media to normal culture media, and incubated them for 18 h. The treatment was repeated three times, and finally we harvested cells at 72 h after the first treatment.

### 2.2. Cell Viability

Thiazoyl blue tetraolium bromide (MTT; Amresco, Solon, OH, USA; product no. 0793) was used to measure cell viability on 24-well plates (SPL Life Science, Pocheon, Gyeonggi, Korea; product no. 30024), as described previously [27,28,29]. Briefly, after discarding the culture medium, cells were incubated with 300 μL of 5 mg/mL MTT in phosphate-buffered saline (PBS; Biosesang; product no. PR4007-100-74) per well for 30 min at 37 °C in 5% CO_2_ incubator. After removing the MTT solution, 300 μL of DMSO was added to each well and incubated for 5 min. To quantify purple-colored formazan product, the absorbance of 100 μL of the incubated DMSO was measured on 96-well plates (SPL Life Science) at 595 nm with a reference wavelength of 620 nm using SpectraMax i3x multimode microplate reader (Molecular Devices, San Jose, CA, USA, in the Bio-Health Materials Core-Facility, Jeju National University). The percentage of cell viability relative to untreated cells was calculated from the average absorbance values of 3 or 4 wells per group. A 50% cytotoxicity concentration (CC_50_) was calculated by nonlinear regression analysis using GraphPad Prism 5.0 (GraphPad Software, San Diego, CA, USA).

### 2.3. Western Blot Analysis

Electrophoresis of protein in cell lysate was performed on 12% or 7.5% polyacrylamide gels prepared from TGX FastCast acrylamide kit (Bio-Rad Laboratories, Hercules, CA, USA; product no. 1610175 or 1610171) in running buffer (Bio-Rad Laboratories; product no. TR2015-100-00) and subsequently transferred to polyvinylidene fluoride membrane (Amersham; product no. 10600021) in transfer buffer (Bio-Rad Laboratories; product no. TR2028-100-00), as described previously [30,31,32]. After blocking the membrane in 5% skim milk (KisanBio, Seoul, Korea; product no. MB-S1667) in tris-buffered saline (TBS; Biosesang; product no. TR2005-100-74) with 0.1% Tween 20 (Biosesang; product no. TR1027-500-00) for 1 h, the membrane was incubated for 18 h at 4 °C with antibodies against α-SMA (1:5000 dilution; Sigma-Aldrich; product no. A5228), fibronectin (1:2500 dilution; ABclonal, Woburn, MA, USA; product no. A12932), p21 (1:2500 dilution; Santa Cruz Biotechnology, Santa Cruz, CA, USA; product no. sc-6246), lamin B1 (1:10,000 dilution; Proteintech; product no. 12987-1-AP), and β-actin (1:5000 dilution; Santa Cruz Biotechnology; product no. sc-47778). After washing in TBS with 0.1% Tween 20, peroxidase anti-rabbit IgG antibody (1:5000 dilution; Vector Laboratories, Burlingame, CA, USA; product no. WB-1000) against fibronectin and lamin B1 antibodies and peroxidase anti-mouse IgG antibody (1:5000 dilution; Vector Laboratories; product no. WB-2000) against α-SMA, p21, and β-actin antibodies were applied for 1 h at room temperature. The protein expression was detected in the Azure c300 imaging system (Azure Biosystems, Dublin, CA, USA) and quantified using the AzureSpot analysis software (Azure Biosystems).

### 2.4. Immunocytochemistry

Immunocytochemistry was performed on 12 mm round coverglass for growth (Thermo Fisher Scientific, Waltham, MA, USA; product no. 12-545-84) in 12-well plates (SPL Life Science; product no. 30012), as previously described [33,34,35]. Briefly, after fixing cells with ice-cold methanol for 5 min, cells were washed thrice with PBS, permeabilizated in 0.1% Triton X-100 (Avantor, Radnor, PA, USA; product no. 0694) for 5 min, blocked in 1% bovine serum albumin (BSA; Bio Basic, Markham, ON, Canada; product no. D0024) for 1 h at room temperature, and stained with α-SMA antibody (1:400 dilution) for 18 h at 4 °C. After three washes with PBS, cells were stained with fluorescein anti-mouse IgG antibody (1:400 dilution; Vector Laboratories; product no. FI-2001) for 1 h at room temperature under protection from light, counterstained with 1 μg/mL 4′-6-diamidino-2-phenylindole dihydrochloride (DAPI; Sigma-Aldrich; product no. D9542) in PBS for 5 min at room temperature, and mounted in VECTASHIELD Antifade mounting medium (Vector Laboratories; product no. H-1000-10). Images were obtained with the Nikon Eclipse Ni microscope, DS-Ri2 camera, and NIS-Elements imaging software, version 4.50, purchased from Nikon (Tokyo, Japan). To evaluate the percentage of α-SMA-positive cells, the numbers of DAPI-positive total cells and α-SMA-positive cells were counted on five randomly chosen low-power fields (×200 magnification) per experiment in a blind manner.

### 2.5. Cell Size

Attachment surface area and nuclear size were examined using staining for F-actin and DNA, respectively, as described previously [36,37]. Briefly, cells were grown on 12 mm round coverglass for growth in 12-well plates, fixed in 4% paraformaldehyde (BioPrince, Chuncheon, Gangwan, Korea; product no. BPP-9016) in PBS for 10 min at room temperature, washed thrice with PBS, stained with 1 μg/mL DAPI in PBS for 5 min at room temperature to determine nuclear boundaries, and mounted in VECTASHIELD Antifade mounting medium with tetramethylrhodamine (TRITC)–phalloidin (Vector Laboratories; product no. H-1600-10) to determine cell boundaries. Images were obtained with the Nikon Eclipse Ni microscope, DS-Ri2 camera, and NIS-Elements imaging software. Cell and nuclear boundaries were detected and measured on five randomly chosen cells per low-power field (×200 magnification) and five randomly chosen low-power fields per experiment in a blind manner. Cell volume was represented by mean value of forward scatter area (FSC-A) per normalized cell number (10,000 events per experiment) using flow cytometry analysis, as previously described [38]. Briefly, cultured adherent cells were detached from 60 mm dishes (SPL Life Science; product no. 20060) with 0.05% trypsin-EDTA, washed with PBS, isolated into a single-cell population, and gated in FSC-A using the CytoFLEX flow cytometer and CytExpert software purchased from Beckman Coulter (Indianapolis, IN, USA). The single-cell population was defined by side scatter height (SSC-H) versus side scatter width (SSC-W) followed by forward scatter height (FSC-H) versus forward scatter width (FSC-W).

### 2.6. Senescence-Associated β-Galactosidase (SA-β-Gal) Activity

To detect cellular senescence, SA-β-gal activity was performed on 60 mm dishes, as modified and described previously [39]. After cell culture, cells were washed with Dulbecco’s PBS excluding MgCl_2_ (Welgene; product no. LB 001-02), fixed in 0.5% glutaraldehyde (Daejung Chemical & Metals, Siheung, Gyeonggi, Korea; product no. 4133-1405) diluted in pure water for 20 min under protection from light, washed with PBS including 100 mM MgCl_2_ (Daejung Chemical & Metals, product no. 5504-4405), and incubated in fresh β-galactosidase staining solution at 37 °C for 48 h. The β-galactosidase staining solution was mixed with 1 mg of 5-bromo-4-chloro-3-indolyl β-D-galactopyranoside (Sigma-Aldrich; product no. 11680893001) per ml dimethylformamide (Biosesang; product no. D1021); 10 mM citric acid (Amresco; product no. 0529) plus 50 mM tri-sodium citrate dihydrate (Sigma-Aldrich; product no. 106448), pH 6.0; 250 mM NaCl (Daejung Chemical & Metals; product no. 7548-4400); 2 mM MgCl_2_; 4 mM potassium ferricyanide (Daejung Chemical & Metals; product no. 6574-4405); and 4 mM potassium hexacyanoferrate(II) trihydrate (Sigma-Aldrich; product no. P9387). After removing the solution, cells were washed with pure water. Images were obtained with the Olympus IX70 inverted microscope equipped with a digital camera and DP Controller software purchased from Olympus (Tokyo, Japan). To evaluate the percentage of SA-β-gal-positive cells, the numbers of total cells and SA-β-gal-positive cells were counted on five randomly chosen low-power fields (×200 magnification) per experiment in a blind manner.

### 2.7. Cell Cycle Assessment

For cell cycle analysis, nuclear DNA was stained with propidium iodide (Sigma-Aldrich; product no. P4170) followed by flow cytometric determination of the DNA content of single cells, as described previously [35,40]. Briefly, cultured adherent cells were detached from 60 mm dishes with 0.05% trypsin-EDTA, washed with PBS, and fixed in ice-cold 70% ethanol (Daejung Chemical & Metals; product no. 4023-4110). After washing cells twice with PBS, cells were incubated in 10 μg of ribonuclease A (Sigma-Aldrich; product no. R4642) plus 22.5 μg of propidium iodide per ml PBS for 15 min at room temperature under protection from light, washed with PBS, isolated into the single-cell population, and gated in propidium iodide area versus height using the CytoFLEX flow cytometer and CytExpert software. Quantitation of cell numbers in cell cycle phases was performed using Modfit LT 5.0 software (Verity Software House, Topsham, ME, USA).

### 2.8. Statistical Analysis

To analyze all data statistically, we used SigmaPlot 14.0 (Systat Software, San Jose, CA, USA) [41]. Briefly, normal distribution was evaluated with Shapiro–Wilk normality test. If this failed, logarithmic transformation was attempted. The parametric data were analyzed with one- or two-way analysis of variance (ANOVA) followed by Tukey’s post hoc test. In F_α,β_ = γ for ANOVA, α, β, and γ denote degree of freedom for explained variance, degree of freedom for residual variance, and F value, respectively. If the logarithmically transformed data also failed, the non-parametric data were analyzed with Kruskal–Wallis test followed by Student–Newman–Keuls post hoc test. In H = α, N_β_ = γ for Kruskal–Wallis test, α, β, and γ denote H value, group number, and sample size, respectively. In figures, parametric data are presented as mean ± standard error of the mean (SEM) with individual data points, and non-parametric data are presented as median value with quartiles. Flow cytometric data on cell cycle phases are presented as mean only. A value of *p* < 0.05 was considered statistically significant. All raw numeric data and statistical results are provided in Appendix A.

## 3. Results

### 3.1. RAC Transforms Kidney Fibroblasts into Myofibroblasts

To establish an in vitro model of RAC in resident kidney fibroblasts, we first evaluated the CC_50_ value of RAC for the NRK-49F cells. Because the CC_50_ value of RAC was 11.89 uM (Figure 1B), the kidney fibroblast cells were exposed to RAC at a final concentration of 0, 5, 10, or 20 μM. To determine whether RAC transforms resident kidney fibroblasts into myofibroblasts, we next measured expression of α-SMA protein in the cells exposed to RAC, as it is the main marker of myofibroblasts. RAC significantly upregulated α-SMA expression at any concentration compared to when it was not used, as determined by Western blot analysis (Figure 1C,D). We also measured the expression of fibronectin protein, as it is a ubiquitous extracellular matrix component. As a result of Western blot analysis, RAC with concentrations of 5 and 10 μM was found to significantly increase fibronectin expression in treated cells when compared to untreated cells, but RAC at a 20 μM concentration did not change that (Figure 1C,E). To complement the Western blot data, we performed immunofluorescence staining of α-SMA in the cells exposed to RAC, and then counted the number of total and α-SMA-positive cells under a low-power fluorescent microscope. RAC-exposed cells showed a marked increase in the percentage of α-SMA-positive cells compared to the percentage seen in untreated cells (Figure 1F,G). Although the percentage of α-SMA-positive cells after exposure to RAC at 20 μM was as high as that in cells exposed to RAC at 10 μM (Figure 1G), RAC at a 20 μM concentration tended to lower α-SMA expression (Figure 1D) and significantly diminished fibronectin expression when compared at a 10 μM concentration (Figure 1E). The cells exposed to RAC at 20 μM had a fall in viability of 80% (Figure 1B), indicating more involvement in cell death induced by RAC than myofibroblast transformation. Because of that, this current study used RAC with a final concentration of 10 μM near the CC_50_ value. These data indicate that RAC induces transformation of resident kidney fibroblasts into myofibroblasts.

### 3.2. RAC Induces Cellular Hypertrophy and Senescence in Kidney Fibroblast Cells

During a transformation into myofibroblasts, fibroblasts derived from lung [42], skin [43], and heart [44] appear to increase in cell size. To analyze the cellular hypertrophy of resident kidney fibroblasts, we used several approaches. We assessed cell attachment surface area and nuclear size in the NRK-49F cells exposed to RAC, as detected by immunofluorescence staining of the cytoarchitectural boundaries of the plasma membrane and nucleus with TRITC-phalloidin and DAPI, respectively. RAC led to a markedly increased cell attachment surface area and nuclear size when compared to those in untreated cells (Figure 2A–C). Because the forward scatter signal from flow cytometer can be converted into cell volume [38], we complemented the microscopic experiments with flow cytometry analysis of FSC-A in the cells exposed to RAC. As a result of increased FSC-A, when compared to untreated cells, RAC significantly enlarged cell volume (Figure 2D,E). These data indicate that RAC induces cellular hypertrophy in resident kidney fibroblast cells.

Since increased cell size is the main feature of cellular senescence [45], the NRK-49F cells exposed to RAC were stained with SA-β-gal to investigate whether RAC induces cellular senescence of resident kidney fibroblasts during the transformation into myofibroblasts. When we counted the number of total and SA-β-gal-positive cells under a low-power light microscope, the percentage of SA-β-gal-positive cells was significantly increased after RAC when compared to untreated cells (Figure 3A,B). Because no single marker for senescence is completely specific, we additionally investigated protein expressions related with cellular senescence. Cellular senescence in the RAC-exposed cells was also confirmed by the upregulated p21 protein and downregulated lamin B1, as determined by Western blot analysis (Figure 3C,E). These data indicate that RAC induces cellular senescence in resident kidney fibroblast cells.

### 3.3. Pharmacological Inhibition of p21 Attenuates RAC-Induced Myofibroblast Transformation in Kidney Fibroblast Cells through Reduction in Cellular Senescence

Upregulation of p21 can mediate cellular senescence in various cell types, including lung cancer cells [46], fibrosarcoma cells [47], and tendon-specific fibroblasts [48]. To determine whether cellular senescence of resident kidney fibroblast cells is responsible for the transformation into myofibroblasts, we used UC2288, a p21 inhibitor, to reduce cellular senescence in the NRK-49F cells exposed to RAC (Figure 4A). When treated with the p21 inhibitor at final concentrations of 0, 1, or 3 μM, there was no significant difference in cell viability between inhibitor-treated and inhibitor-untreated cells (Figure 4B), indicating no cytotoxicity of the p21 inhibitor for the resident kidney fibroblast cells. As determined by Western blot analysis, treatment with the p21 inhibitor at a final concentration of 3 μM efficaciously reduced p21 expression in the RAC-exposed cells (Figure 4C,D), together with the upregulation of lamin B1 protein (Figure 4C,E) compared to inhibitor-untreated cells. In addition, pharmacological inhibition of p21 expression in the RAC-exposed cells significantly decreased the percentage of SA-β-gal-positive cells compared to that of inhibitor-untreated cells (Figure 4F,G). These data indicate that p21 inhibition attenuates cellular senescence induced by RAC in resident kidney fibroblast cells.

The microscopic study showed that the inhibition of p21 expression markedly reduced the cell attachment surface area (Figure 5A,B) and nuclear size (Figure 5A,C) of treated cells when compared to those of inhibitor-untreated cells, indicating a reduction in cellular hypertrophy of resident kidney fibroblast cells by p21 inhibition. Furthermore, Western blot analysis demonstrated that the p21 inhibition resulted in a marked downregulation of α-SMA in the cells exposed to RAC, when compared to inhibitor-untreated cells (Figure 5D,E), indicating a reduction in the transformation of resident kidney fibroblasts into myofibroblasts. These results suggest that p21-dependent cellular senescence contributes to myofibroblast transformation after RAC in resident kidney fibroblast cells.

### 3.4. G2/M Cell Cycle Arrest Contributes to Cellular Hypertrophy, Senescence, and Myofibroblast Transformation in Kidney Fibroblasts

As demonstrated by flow cytometry analysis (Figure 6A,B), RAC-exposed resident kidney fibroblast cells showed a significant and dose-dependent increase in the percentage of cells in the G2/M phase from approximately 12%, 27%, 41%, and 60% in RAC at concentrations of 0, 5, 10, and 20 μM, respectively. In contrast, the percentage of cells in the G0/G1 phase was significantly and dose-dependently reduced in the RAC-exposed cells compared to untreated cells. These results indicate that RAC induces G2/M cell cycle arrest in resident kidney fibroblast cells.

Under conditions of tissue injury, prolonged cell cycle arrest at the G2/M phase contributes to myofibroblast transformation in hepatic stellate cells [49], corneal endothelial cells [50], and kidney tubular epithelial cells [51]. To verify whether repeated arrest at the G2/M boundary leads to consequences of RAC in resident kidney fibroblast cells, we used paclitaxel, a microtubule-interfering agent, to induce progressive G2/M cell cycle arrest through the prevention of mitotic spindle formation [26] in the NRK-49F cells, and carried out the procedure according to the timetable in Figure 7A. Repeated administration of paclitaxel (RAP) with final concentrations of 0 to 10 nM significantly reduced cell viability in treated cells compared to untreated cells (Figure 7B), indicating that low-dose paclitaxel inhibits cell proliferation of resident kidney fibroblast cells, but may not induce cytotoxicity. The result obtained from flow cytometry analysis showed that RAP at a final concentration of 10 nM efficaciously heightened the percentage of cells in the G2/M phase to approximately 30% when compared to the percentage of untreated cells (Figure 7C,D). Coincident with RAC, we found that RAP significantly increased cell attachment surface area (Figure 8A,B), nuclear size (Figure 8A,C), and cell volume (Figure 8D,E) in treated cells compared to untreated cells, indicating that RAP induces cellular hypertrophy in resident kidney fibroblast cells. In addition, cellular senescence in the cells was induced by RAP, as demonstrated by marked increments in the percentage of SA-β-gal-positive cells (Figure 8F,G) and p21 expression (Figure 8H,I), and a significant decrement in lamin B1 expression (Figure 8H,J) in RAP-treated cells when compared to untreated cells. Finally, we measured α-SMA and fibronectin expression in the RAP-exposed cells to investigate whether G2/M arrest transforms resident kidney fibroblast cells into myofibroblasts. The results obtained from Western blot analysis show that RAP significantly upregulated α-SMA and fibronectin (Figure 8H,K,L), indicating RAP-induced transformation of resident kidney fibroblast cells into active myofibroblasts. Taken together, these results suggest that G2/M cell cycle arrest is responsible for cellular senescence and myofibroblast transformation in resident kidney fibroblast cells.

## 4. Discussion

Cancer patients are repeatedly administered low doses of cisplatin over a long period of time, and in repeated episodes of cisplatin-induced AKI, tubulointerstitial fibrosis can occur [52]. In pathogenetic studies of cisplatin-induced tubulointerstitial fibrosis, a resident kidney fibroblast is a poor fibrogenic candidate, irrespective of the key contributor to the AKI-to-CKD progression [53]. In the current study, we have shown, for the first time to our knowledge, that (1) RAC transforms resident kidney fibroblasts into myofibroblasts; (2) RAC induces cellular hypertrophy and senescence in resident kidney fibroblast cells; (3) p21-dependent cellular senescence contributes to the fibroblast-to-myofibroblast transformation induced by RAC; and (4) G2/M cell cycle arrest in resident kidney fibroblasts causes cellular hypertrophy, senescence, and finally transformation into myofibroblasts (Figure 9).

Transformation of fibroblasts into myofibroblasts is a common pathological phenomenon of chronic disease including CKD [54]. The transformed myofibroblast is characterized by upregulation of α-SMA with a stress-fiber-like appearance that is involved in the production of extracellular matrix components such as collagen and fibronectin [55]. RAC enhances α-SMA expression in whole mouse kidneys [8,9]. Here, we showed that RAC dramatically upregulated α-SMA protein in resident kidney fibroblasts. Our results suggest that myofibroblasts transformed from kidney fibroblasts are responsible for α-SMA expression in fibrotic kidneys induced by RAC. Similar to our result that RAC enhanced fibronectin expression in resident kidney fibroblasts, repeated injury to mouse kidney proximal tubular cells using cisplatin [9,22,23] and simian diphtheria toxin [56] upregulates fibronectin protein and profibrotic mRNA in the tubule epithelial cells. The results of other investigators suggest that consistent with resident kidney fibroblasts, kidney tubule epithelial cells also directly produce extracellular matrix components after repeated injury, resulting in a contribution to tubulointerstitial fibrosis. Whether resident kidney fibroblasts are a major contributor to RAC-induced tubulointerstitial fibrosis compared with tubular epithelial cells remains to be explored.

A single administration of cisplatin at a high dose induces cell death, whereas RAC at a low dose may change sublethal properties instead of lead to cell death [57]. Like kidney tubule epithelial cells with high susceptibility to cisplatin, skin fibroblasts also undergo DNA damage and apoptotic cell death after a single administration of a high dose of cisplatin [58,59]. In our current study, resident kidney fibroblasts had senescent properties after RAC at a low dose, as demonstrated by a combination of three markers, including an increased percentage of SA-β-gal-positive cells, upregulation of p21 protein, and downregulation of lamin B1 protein. SA-β-gal activity based on lysosomal β-galactosidase gene upregulation is the most extensively used marker for senescent cells [60]. Cyclin-dependent kinase inhibitor p21 and nuclear lamina protein lamin B1 are upregulated and downregulated in response to senescence-inducing stimuli, respectively [47,61]. p21 especially is a critical suppressor of the proliferating cell nuclear antigen DNA replication factor through inactivation of both cyclin E- and D1-associated kinases during the early stage of senescence [62]. In the current study, RAC markedly upregulated p21 protein, and pharmacological inhibition of p21 expression significantly limited the decrement in lamin B1 expression and increment in the percentage of SA-β-gal-positive cells induced by RAC, suggesting that RAC-induced p21 upregulation mediates cellular senescence in resident kidney fibroblast cells. In addition, p21 inhibition efficaciously attenuated transformation of resident kidney fibroblasts into myofibroblasts, as demonstrated by the marked downregulation of α-SMA protein. Consistent with our results, p21 knockdown in human skin and lung fibroblasts disrupts maintenance of DNA-damage-induced senescence [63]. Furthermore, p21 deficiency in mice attenuates SA-β-gal activity and α-SMA expression in carbon-tetrachloride-induced liver fibrosis [63] and limits tubulointerstitial fibrosis in diabetic nephropathy [64]. Our research and other studies suggest that p21-dependent senescence in resident kidney fibroblasts contributes to tubulointerstitial fibrosis. An aberrant increase in cell size is associated with cellular senescence with an inability to engage in cell division [65]. An in vivo examination of 24-month-old mice showed that senescent cells of various types were over 9-fold larger than non-senescent cells [66]. Large hematopoietic stem cells isolated from mice also exhibited the most fundamental of senescent phenotypes, including an inability to proliferate [66]. Among senescence-related proteins, increased expressions of p21 and p16 are associated with cellular hypertrophy. Among those proteins, the increase in p21 expression is caused by an increased amount of DNA damage, possibly because the DNA damage results in size-dependent p53 activation and transcription of its downstream p21 [67,68]. It is well-known that chemical manipulations blocking cell proliferation or cell cycle progression induce an aberrant increase in cell size [69]. For example, treatment of human lung fibroblasts with doxorubicin, a DNA-damaging agent, increased cell volume by approximately 8-fold [70]. Consistent with the above studies, our current findings show that RAC dramatically increased attachment surface area and cell volume in resident kidney fibroblast cells, indicating RAC-induced cellular hypertrophy. Furthermore, pharmacological inhibition of p21 expression significantly diminished the cellular hypertrophy induced by RAC. Our results therefore show that RAC-induced p21 upregulation contributes to cellular hypertrophy in resident kidney fibroblast cells. On the other hand, nuclear size in senescent cells is generally larger than that in normal cells, even though the amount of DNA contents reflects the G1 phase of the cell cycle in the majority of cells [71]. The increase in nuclear size can be caused by downregulated lamin B1, a major component of the nuclear lamina, because its downregulation weakens the nuclear architecture and then increases the nuclear size [72]. Consistently, our results show that RAC enhanced nuclear size and reduced lamin B1 expression in resident kidney fibroblasts. Intriguingly, p21 inhibition in the cells after RAC augmented lamin B1 expression, suggesting that p21 regulates the transactivation of lamin B1. The negative contribution of p21 to lamin B1 in cellular hypertrophy remains to be explored.

An increase in cell size affects alterations in intracellular transport, the surface-to-volume ratio, and the cytoplasm volume-to-DNA content ratio [73]. Reduced ratios of nucleus-to-cytoplasm and volume-to-attachment surface area cause a stretch injury to cells [74,75]. In our current study, quantitative microscopic and cytometric analysis of resident kidney fibroblast after RAC showed that the attachment surface area was approximately 11.0-, 21.9-, and 27.6-fold larger when compared with nuclear sizes with 3.1-, 4.7-, and 4.9-fold increases or when compared with cell volume with 1.3-, 1.7-, and 2.2-fold increases, in a dose-dependent manner, respectively. These calculations indicate that RAC dramatically reduces the ratios of volume to attachment surface area and nucleus to cytoplasm. To maintain architecture of non-muscle cells during mechanical stretch, a stress fiber formation including α-SMA expression is induced, leading to increased resistance to the excessive extent of cytoplasm [76]. Our research and other studies have demonstrated that mechanical stretch upregulates α-SMA protein in resident kidney fibroblasts using Western blot analysis and immunohistochemistry [77,78,79]. Taken together, our research and other findings suggest that cellular stretch of resident kidney fibroblasts occurs due to cellular hypertrophy induced by RAC and causes upregulation of α-SMA protein.

There are a number of previous studies presenting findings that are supportive of our results, which show that G2/M cell cycle arrest transforms resident kidney fibroblasts into myofibroblasts. Previous investigations on the mechanism of cellular senescence have focused on cell cycle blockage in the G1 phase to prevent the initiation of DNA replication in damaged cells [80,81]. Senescent cells can also stop in the G2 phase to block mitosis in response to stress and DNA damage [82,83]. Treatment with cisplatin can induce G2/M cell cycle arrest in human hepatocellular carcinoma cells [84] and non-small-cell lung cancer cells [85]. In our current study, we verified that RAC induced cell cycle arrest at the G2/M phase in resident kidney fibroblasts and proved that RAP-induced G2/M cell cycle arrest led to cellular hypertrophy, senescence including p21 upregulation, and transformation of fibroblasts into myofibroblasts. p21 is a transcriptional target of p53 contributing to the G2 checkpoint and also participates in the G2 checkpoint by inhibiting cyclin-dependent kinases [86,87]. A sustained increase in G2/M arrest induced by paclitaxel in mouse kidneys after mild ischemia and reperfusion injury develops into tubulointerstitial fibrosis [88]. G2/M arrest induced by ultraviolet A irradiation in mouse eyes also reveals cellular senescence, hypertrophy, and upregulation of α-SMA mRNA [50].

In conclusion, we have demonstrated that repeated administration of low-dose of cisplatin transforms resident kidney fibroblasts into myofibroblasts through G2/M cell cycle arrest, p21-dependent cellular hypertrophy, and senescence. This elucidation of the cellular mechanisms underlying the transformation of fibroblasts into myofibroblasts provides valuable information for designing effective therapeutic approaches to prevent or even reverse cisplatin-induced chronic kidney disease. Furthermore, the unique in vitro model developed using resident kidney fibroblast cells from our current study provides a useful tool for the investigation of chronic effects of cisplatin chemotherapy in kidneys.

## Figures and Tables

**Figure 1 cells-11-03472-f001:**
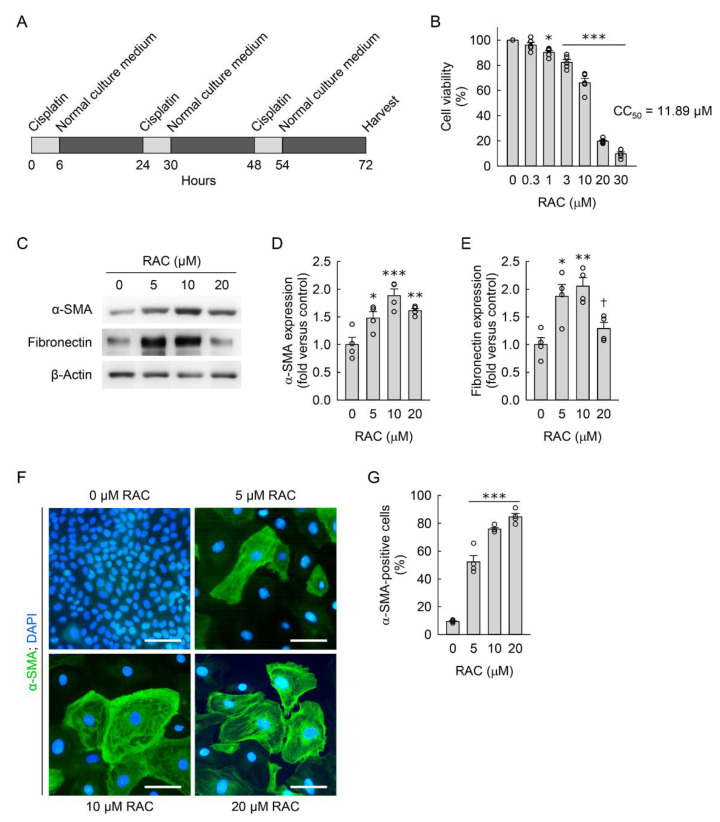
Repeated administration of cisplatin (RAC) induces transformation of kidney fibroblasts into myofibroblasts. (**A**) Experimental timetable for RAC in normal rat kidney fibroblast NRK-49F cells. The cells were treated with 0, 0.3, 1, 3, 5, 10, 20, or 30 μM cisplatin in normal culture media for 6 h, and the media were replaced with normal culture media, followed by incubation for 18 h. The experiment was repeated three times, and finally the cells were harvested at 72 h after the first treatment. (**B**) Cell viability was measured using 3-(4,5-dimethyldiazol-2-yl)-2,5-diphenyltetrazolium bromide dye after RAC (*n* = 5 experiments, 4 wells per experiment). (**C**) Representative Western blots of α-smooth muscle actin (α-SMA) and fibronectin expression. Anti-β-actin antibody was used as a loading control for Western blot analysis. (**D**,**E**) Quantifications of α-SMA and fibronectin expression using the Azure c300 imaging system (Azure Biosystems, Dublin, CA, USA) (*n* = 4 experiments). (**F**) Representative images of α-SMA-stained cells at ×400 magnification using the Nikon Eclipse Ni upright microscope (Nikon, Tokyo, Japan), DS-Ri2 camera (Nikon), and NIS-Elements imaging software (Nikon). Total cellular DNA was stained with 4′,6-diamidino-2-phenylindole (DAPI). Scale bar, 50 μm. (**G**) The percentage of α-SMA-positive cells was counted at ×200 magnification (*n* = 4 experiments). (**B**,**D**,**E**,**G**) Data are presented as mean ± standard error of the mean with individual data points. Statistical significance was determined by one-way analysis of variance followed by Tukey’s post hoc test ((**B**), F_6,28_ = 361.682, *p* < 0.001; (**D**), F_3,12_ = 11.564, *p* < 0.001; (**E**), F_3,12_ = 9.846, *p* = 0.001; (**G**), F_3,12_ = 369.266, *p* < 0.001). * *p* < 0.05, ** *p* < 0.01, *** *p* < 0.001 versus 0 μM RAC; ^†^ *p* < 0.05 versus 10 μM RAC.

**Figure 2 cells-11-03472-f002:**
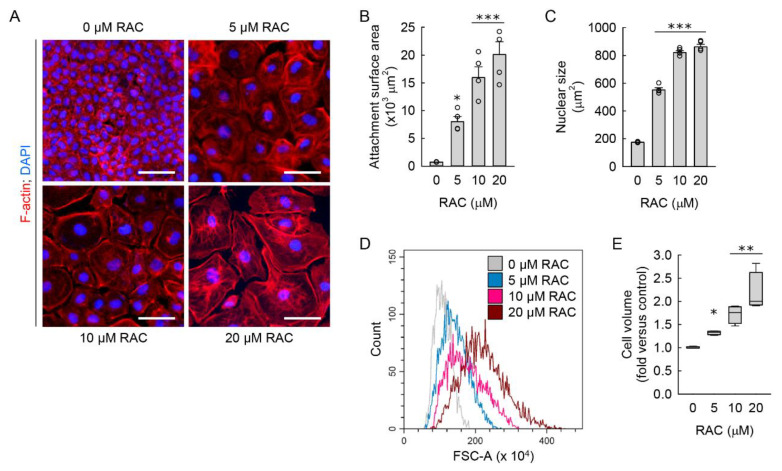
Repeated administration of cisplatin (RAC) induces cellular hypertrophy in kidney fibroblast cells. Normal rat kidney fibroblast NRK-49F cells were treated with 0, 5, 10, or 20 μM cisplatin in normal culture media for 6 h, and the media were replaced with normal culture media, followed by incubation for 18 h. The experiment was repeated three times, and finally, the cells were harvested at 72 h after the first treatment. (**A**) Representative images of F-actin-stained cells at ×400 magnification using the Nikon Eclipse Ni upright microscope (Nikon, Tokyo, Japan), DS-Ri2 camera (Nikon), and NIS-Elements imaging software (Nikon). Total cellular DNA was stained with 4′,6-diamidino-2-phenylindole (DAPI). Scale bar, 50 μm. (**B**,**C**) Quantifications of cell attachment surface area from F-actin staining and cell nuclear size from DAPI staining at ×200 magnification using NIS-Elements imaging software, respectively (*n* = 4 experiments). Data are presented as mean ± standard error of the mean with individual data points. Statistical significance was determined by one-way analysis of variance followed by Tukey’s post hoc test ((**B**) F_3,12_ = 29.303, *p* < 0.001; (**C**) F_3,12_ = 541.88, *p* < 0.001). *** *p* < 0.001 versus 0 μM RAC. (**D**) Representative histogram of forward scatter area (FSC-A) using the CytoFLEX flow cytometer (Beckman Coulter, Indianapolis, IN, USA) and CytExpert software (Beckman Coulter). (**E**) Cell volume was defined based on a mean value of FSC-A per normalized cell number using flow cytometry (*n* = 4 experiments, 10,000 events per experiment). Data are presented as the median value with quartiles. Statistical significance was determined by Kruskal–Wallis followed by Student–Newman–Keuls post hoc test (H = 14.118, N_1–4_ = 4, *p* = 0.003). * *p* < 0.05, ** *p* < 0.01 versus 0 μM RAC.

**Figure 3 cells-11-03472-f003:**
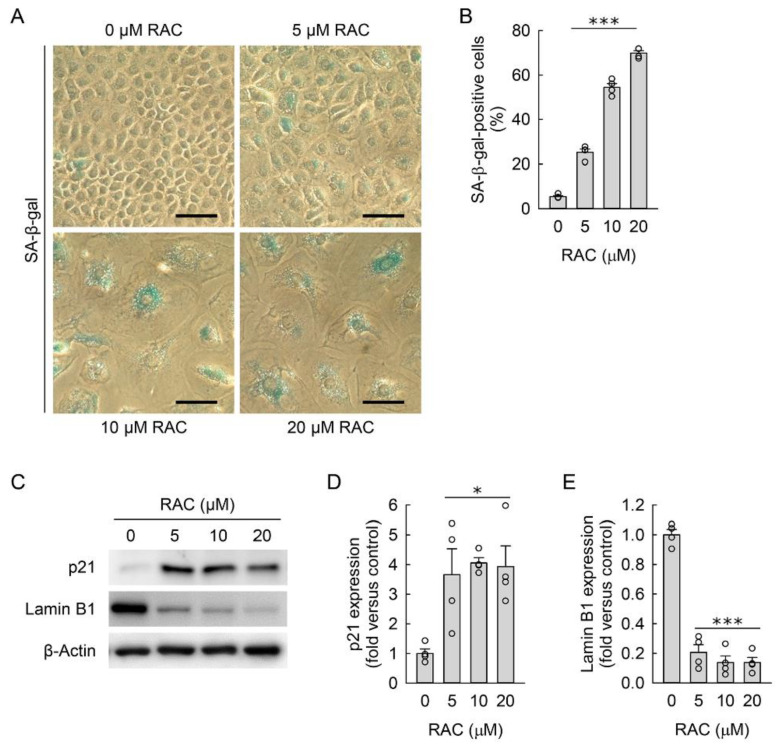
Repeated administration of cisplatin (RAC) induces cellular senescence in kidney fibroblast cells. Normal rat kidney fibroblast NRK-49F cells were treated with 0, 5, 10, or 20 μM cisplatin in normal culture media for 6 h, and the media were replaced with normal culture media, followed by incubation for 18 h. The experiment was repeated three times, and finally the cells were harvested at 72 h after the first treatment. (**A**) Representative images of senescence-associated β-galactosidase (SA-β-gal)-stained cells at ×400 magnification using the Olympus IX70 inverted microscope (Olympus, Tokyo, Japan), DP Controller camera (Olympus), and DPManager imaging software (Olympus). Scale bar, 50 μm. (**B**) The percentage of SA-β-gal-positive cells was counted at ×200 magnification (*n* = 4 experiments). (**C**) Representative Western blots of p21 and lamin B1 expression. Anti-β-actin antibody was used as a loading control of Western blot analysis. (**D**,**E**) Quantifications of p21 and lamin B1 expression using the Azure c300 imaging system (Azure Biosystems, Dublin, CA, USA) (*n* = 4 experiments). (**B**,**D**,**E**) Data are presented as mean ± standard error of the mean with individual data points. Statistical significance was determined by one-way analysis of variance followed by Tukey’s post hoc test ((**B**) F_3,12_ = 496.935, *p* < 0.001; (**D**) F_3,12_ = 6.457, *p* = 0.008; (**E**) F_3,12_ = 98.019, *p* < 0.001). * *p* < 0.05, *** *p* < 0.001 versus 0 μM RAC.

**Figure 4 cells-11-03472-f004:**
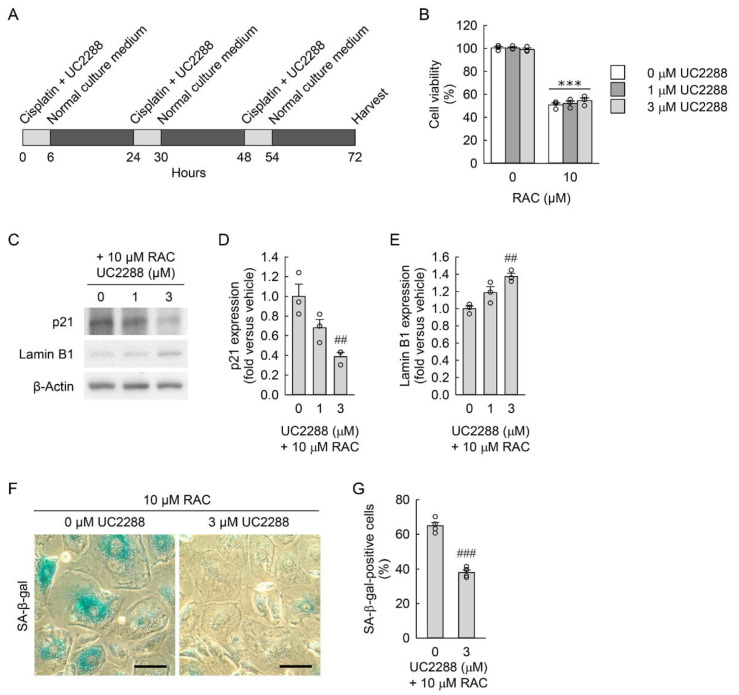
Inhibition of p21 reduces cellular senescence induced by repeated administration of cisplatin (RAC). (**A**) Experimental timetable for RAC and UC2288, a cell-permeable p21 inhibitor, in normal rat kidney fibroblast NRK-49F cells. The cells were treated with 0, 1, or 3 μM UC2288 in 0.1% dimethyl sulfoxide and 10 μM cisplatin in normal culture media for 6 h, and the media were replaced with normal culture media, followed by incubation for 18 h. The experiment was repeated three times, and finally the cells were harvested at 72 h after the first treatment. (**B**) Cell viability was measured using 3-(4,5-dimethyldiazol-2-yl)-2,5-diphenyltetrazolium bromide dye (*n* = 3 experiments, 3 wells per experiment). Data are presented as mean ± standard error of the mean with individual data points. Statistical significance was determined by two-way analysis of variance (ANOVA) followed by Tukey’s post hoc test (effect of RAC, F_1,12_ = 1149.211, *p* < 0.001; effect of UC2288, F_2,12_ = 0.285, *p* = 0.757; interaction between RAC and UC2288, F_2,12_ = 1.188, *p* = 0.338). *** *p* < 0.001 versus 0 μM RAC. (**C**) Representative Western blots of p21 and lamin B1 expression. Anti-β-actin antibody was used as a loading control of Western blot analysis. (**D**,**E**) Quantifications of p21 and lamin B1 expression using the Azure c300 imaging system (Azure Biosystems, Dublin, CA, USA) (*n* = 3 experiments). (**F**) Representative images of senescence-associated β-galactosidase (SA-β-gal)-stained cells at ×400 magnification using the Olympus IX70 inverted microscope (Olympus, Tokyo, Japan), DP Controller camera (Olympus), and DPManager imaging software (Olympus). Scale bar, 50 μm. (**G**) The percentage of SA-β-gal-positive cells was counted at ×200 magnification (*n* = 4 experiments). (**D**,**E**,**G**) Data are presented as mean ± standard error of the mean with individual data points. Statistical significance was determined by ANOVA followed by Tukey’s post hoc test ((**D**), F_2,6_ = 11.535, *p* = 0.009; (**E**) F_2,6_ = 14.163, *p* = 0.005; (**G**) F_1,6_ = 112.567, *p* < 0.001). ## *p* < 0.01, ### *p* < 0.001 versus 0 μM UC2288.

**Figure 5 cells-11-03472-f005:**
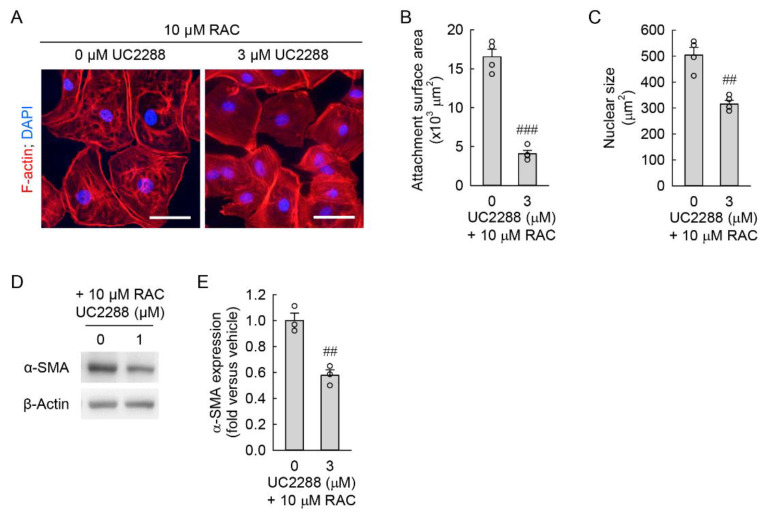
Inhibition of p21 reduces cellular hypertrophy and myofibroblast transformation induced by repeated administration of cisplatin (RAC). Normal rat kidney fibroblast NRK-49F cells were treated with 0, 1, or 3 μM UC2288, a cell-permeable p21 inhibitor, in 0.1% dimethyl sulfoxide and 10 μM cisplatin in normal culture media for 6 h, and the media were replaced with normal culture media, followed by incubation for 18 h. The experiment was repeated three times, and finally, the cells were harvested at 72 h after the first treatment. (**A**) Representative images of F-actin-stained cells at ×400 magnification using the Nikon Eclipse Ni upright microscope (Nikon, Tokyo, Japan), DS-Ri2 camera (Nikon), and NIS-Elements imaging software (Nikon). Total cellular DNA was stained with 4′,6-diamidino-2-phenylindole (DAPI). Scale bar, 50 μm. (**B**,**C**) Quantification of cell attachment surface area from F-actin staining and cell nuclear size from 4′,6-diamidino-2-phenylindole staining at ×200 magnification using NIS-Elements imaging software (Nikon, Tokyo, Japan) (*n* = 4 experiments). (**D**) Representative Western blot of α-smooth muscle actin (α-SMA) expression. Anti-β-actin antibody was used as a loading control of Western blot analysis. (**E**) Quantification of α-SMA expression using the Azure c300 imaging system (*n* = 4 experiments). (**B**,**C**,**E**) Data are presented as mean ± standard error of the mean with individual data points. Statistical significance was determined by ANOVA followed by Tukey’s post hoc test ((**B**) F_1,6_ = 134.557, *p* < 0.001; (**C**) F_1,6_ = 31.504, *p* = 0.001; (**E**) F_1,4_ = 34.633, *p* = 0.004). ## *p* < 0.01, ### *p* < 0.001 versus 0 μM UC2288.

**Figure 6 cells-11-03472-f006:**
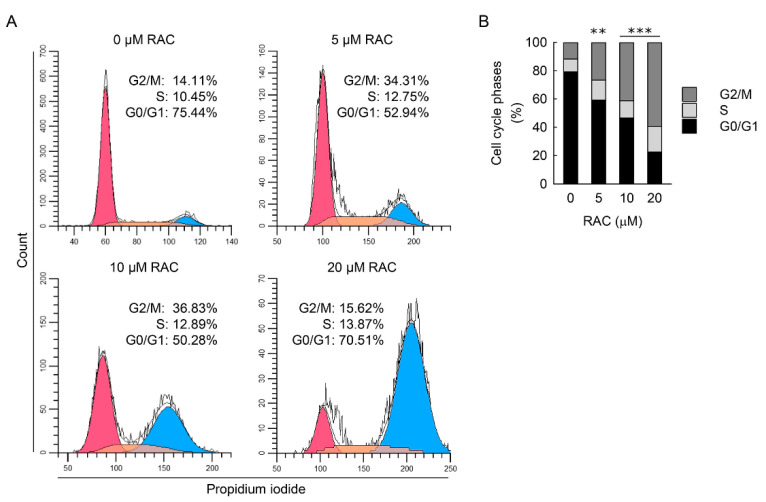
Repeated administration of cisplatin (RAC) leads to G2/M cell cycle arrest in kidney fibroblast cells. Normal rat kidney fibroblast NRK-49F cells were treated with 0, 5, 10, or 20 μM cisplatin in normal culture media for 6 h, and the media were replaced with normal culture media, followed by incubation for 18 h. The experiment was repeated three times, and finally, the cells were harvested at 72 h after the first treatment. (**A**) Representative histogram of cell cycle using the CytoFLEX flow cytometer (Beckman Coulter, Indianapolis, IN, USA), CytExpert software (Beckman Coulter), and Modfit LT 5.0 software (Verity Software House, Topsham, ME, USA). (**B**) Quantification of cell cycle phases using Modfit LT 5.0 software (*n* = 4 experiments). Data are presented as mean of cell cycle phases. Statistical significance was determined by two-way analysis of variance (ANOVA) followed by Tukey’s post hoc test (effect of RAC, F_3,48_ = 0.0000208, *p* = 1.000; effect of cell cycle, F_2,48_ = 146.699, *p* < 0.001; interaction between RAC and cell cycle, F_6,48_ = 49.164, *p* < 0.001). ** *p* < 0.01, *** *p* < 0.001 versus 0 μM RAC.

**Figure 7 cells-11-03472-f007:**
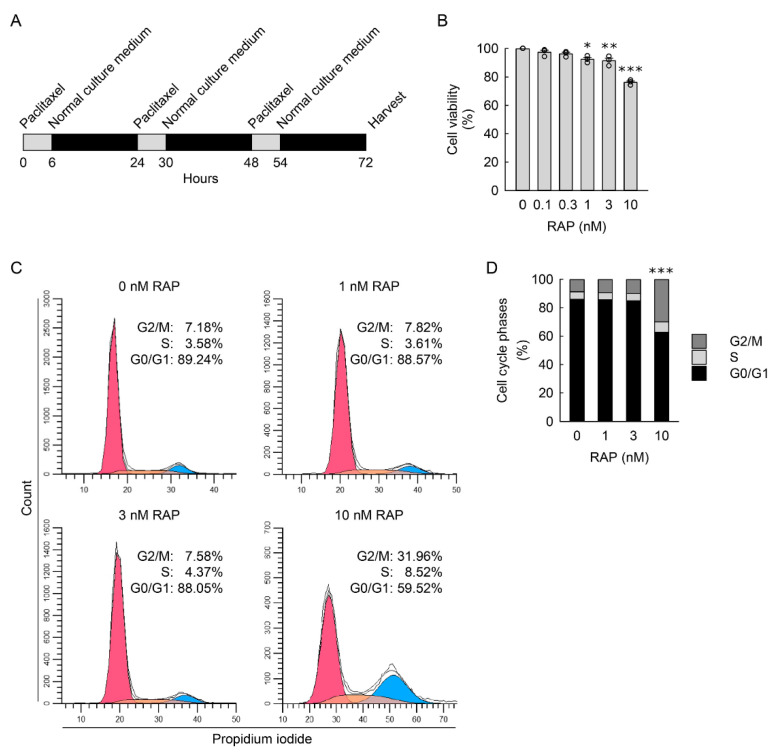
Repeated administration of paclitaxel (RAP) induces G2/M cell cycle arrest. (**A**) Experimental timetable for RAP in normal rat kidney fibroblast NRK-49F cells. The cells were treated with 0, 0.1, 0.3, 1, 3, or 10 nM paclitaxel, a microtubule-stabilizing agent, in normal culture media for 6 h, and the media were replaced with normal culture media, followed by incubation for 18 h. The experiment was repeated three times, and finally, the cells were harvested at 72 h after the first treatment. (**B**) Cell viability was measured using 3-(4,5-dimethyldiazol-2-yl)-2,5-diphenyltetrazolium bromide dye (*n* = 3 experiments, 4 wells per experiment). Data are presented as mean ± standard error of the mean with individual data points. Statistical significance was determined by one-way analysis of variance (ANOVA) followed by Tukey’s post hoc test (F_5,12_ = 42.102, *p* < 0.001). * *p* < 0.05, ** *p* < 0.01, *** *p* < 0.001 versus 0 nM RAP. (**C**) Representative histogram of cell cycle using the CytoFLEX flow cytometer (Beckman Coulter, Indianapolis, IN, USA), CytExpert software (Beckman Coulter), and Modfit LT 5.0 software (Verity Software House, Topsham, ME, USA). (**D**) Quantification of cell cycle phases using the CytoFLEX flow cytometer (Beckman Coulter, Indianapolis, IN, USA), CytExpert software (Beckman Coulter), and Modfit LT 5.0 software (Verity Software House, Topsham, ME, USA) (*n* = 4 experiments). Data are presented as mean of cell cycle phases. *** *p* < 0.001 versus 0 nM RAP. Statistical significance was determined by two-way ANOVA followed by Tukey’s post hoc test (effect of RAP, F_3,36_ = 0.112, *p* = 0.953; effect of cell cycle, F_2,36_ = 5366.562, *p* < 0.001; interaction between RAP and cell cycle, F_6,36_ = 99.768, *p* < 0.001).

**Figure 8 cells-11-03472-f008:**
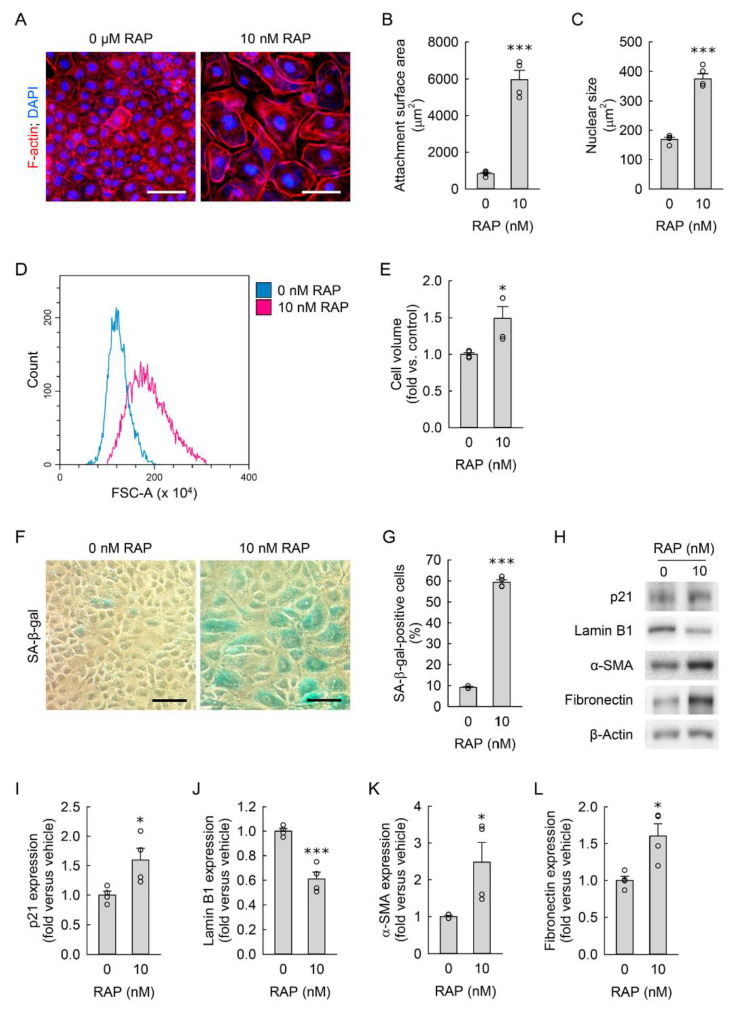
Repeated administration of paclitaxel (RAP) induces cellular hypertrophy and cellular senescence within transformation of kidney fibroblasts into myofibroblasts. Normal rat kidney fibroblast NRK-49F cells were treated with 0 or 10 nM paclitaxel, a microtubule-stabilizing agent, in normal culture media for 6 h, and the media were replaced with normal culture media, followed by incubation for 18 h. The experiment was repeated three times, and finally, the cells were harvested at 72 h after first treatment. (**A**) Representative images of F-actin-stained cells at ×400 magnification using the Nikon Eclipse Ni upright microscope (Nikon, Tokyo, Japan), DS-Ri2 camera (Nikon), and NIS-Elements imaging software (Nikon). Total cellular DNA was stained with 4′,6-diamidino-2-phenylindole (DAPI). Scale bar, 50 μm. (**B**,**C**) Quantifications of cell attachment surface area from F-actin staining and cell nuclear size from DAPI staining at ×200 magnification using NIS-Elements imaging software (*n* = 4 experiments). (**D**) Representative histogram of forward scatter area (FSC-A) using the CytoFLEX flow cytometer (Beckman Coulter, Indianapolis, IN, USA) and CytExpert software (Beckman Coulter). (**E**) Cell volume was defined based on a mean value of forward scatter area using the CytoFLEX flow cytometer and CytExpert software (*n* = 4 experiments, 10,000 events per experiment). (**F**) Representative images of senescence-associated β-galactosidase (SA-β-gal)-stained cells at ×400 magnification using the Olympus IX70 inverted microscope (Olympus, Tokyo, Japan), DP Controller camera (Olympus), and DPManager imaging software (Olympus). Scale bar, 50 μm. (**G**) The percentage of SA-β-gal-positive cells was counted at ×200 magnification (*n* = 4 experiments). (**H**) Representative Western blots of p21, lamin B1, α-smooth muscle actin (α-SMA), and fibronectin expression. Anti-β-actin antibody was used as a loading control of Western blot analysis. (**I**–**L**) Quantifications of p21, lamin B1, α-smooth muscle actin (α-SMA), and fibronectin expression using the Azure c300 imaging system (Azure Biosystems, Dublin, CA, USA) (*n* = 4 experiments). (**B**,**C**,**E**,**G**,**I**–**L**) Data are presented as mean ± standard error of the mean with individual data points. Statistical significance was determined by one-way ANOVA followed by Tukey’s post hoc test ((**B**) F_1,6_ = 107.212, *p* < 0.001; (**C**) F_1,6_ = 131.068, *p* < 0.001; (**E**) F_1,6_ = 9.864, *p* = 0.020; (**G**) F_1,6_ = 1684.723, *p* < 0.001; (**I**) F_1,6_ = 7.794, *p* = 0.032; (**J**) F_1,6_ = 42.759, *p* < 0.001; (**K**) F_1,6_= 7.293, *p* = 0.036; (**L**) F_1,6_ = 11.762, *p* = 0.014). * *p* < 0.05, *** *p* < 0.001 versus 0 nM RAP.

**Figure 9 cells-11-03472-f009:**
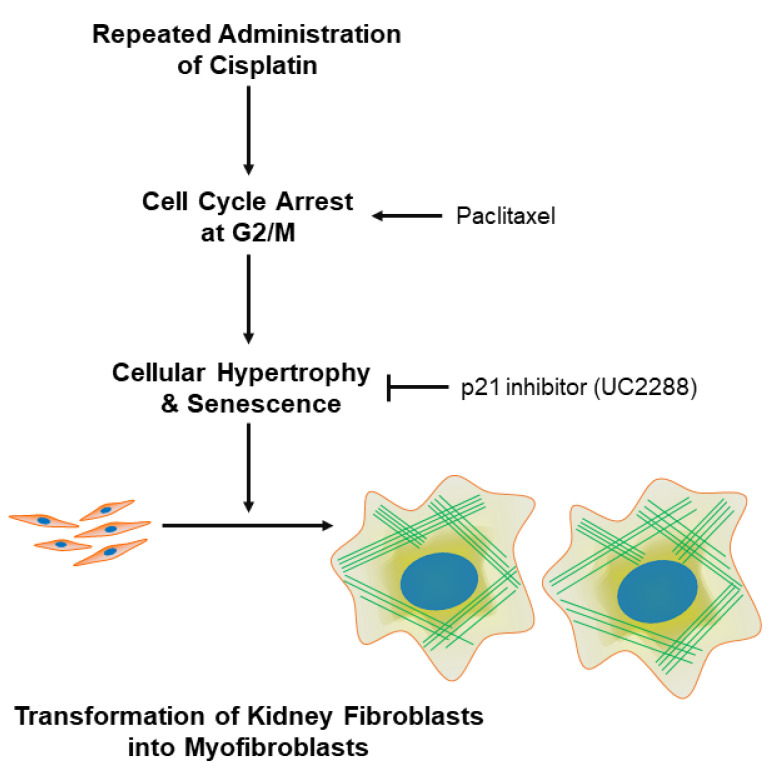
Scheme of mechanism of kidney fibroblast-to-myofibroblast transformation during exposure to RAC.

## Data Availability

The data presented in this study are available from the corresponding author upon reasonable request.

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
