# Peer review of "Repeated Administration of Cisplatin Transforms Kidney Fibroblasts through G2/M Arrest and Cellular Senescence"

_cells, 2022, doi:10.3390/cells11213472_

Round 1

Reviewer 1 Report

Thank you so much for giving me an opportunity to review an interesting research paper and work done by Jia-Bin Yu etal. 

Authors have presented interesting subject on how a chemotherapeutic agent cisplatin transforms kidney fibroblasts through G2/M arrest and cellular senescence. 

My best wishes to the authors for future work and studies.

Reviewer 2 Report

This article is about the treatment effect of 0-30 μM cisplatin for six h (3 times) on the rat kidney fibroblast cell line (NRK-49F). The authors demonstrate that a three-day treatment with low concentrations of cisplatin transforms fibroblasts into myofibroblasts. The manuscript characterizes the changes that the cell line had; however, it is necessary to highlight the importance and impact that this finding could have further. It is an interesting topic, but I consider that before its publication, the following aspects are addressed:

Principal changes

These changes should add to the text.

Methodology

1.      Please clarify why the cells are only starved for two hours, since it is usually starved all night.

Results

2.      In the figures, the control group is abbreviated as con, however, it is confusing. I suggest that the control group appears as 0 (zero).

3.      Fig 2B shows the result with 5μM RAC; however, Figure 2A omits these results, why? The same happens with Figures 3A and 3B

4.      Figure 4. Please clarify if 10 μM RAC was used for all assays (A-J).

5.      Caption 4. F: The image does not appear, only the quantification.

Discussion

6.      Although poorly mentioned in lines 576 to 577; the authors should highlight the importance of their results.

7.      Fig 1B. The experiment lacks a control group, which is essential to compare the change in cell viability of the study group. The control group is mentioned in the methodology, but it does not appear in the figure.

8.      Figure 1B. Although it is a logarithm scale, the difference in cell viability between 10 μM (70%) and 30 μM (10%) is very large; I suggest including an intermediate point, such as 20 μM.

9.      Figure 1. Explain why the expression of α-SMA and fibronectin decreased at 20 μM RAC.

10.   There are other articles that use low concentrations of CP repeatedly in other cell lines, so it would be important to compare these findings with their results.

11.   Add to the discussion if cisplatin treatment in other cell lines tends to arrest the cell cycle in those same phases (G2/M).

12.   The authors must inform about the limitations of the research.

13.   The concentrations used in this study (5, 10 and 20 μM RAC) should be compared with those used in in vitro model of kidney damage induced by cisplatin (nephrotoxicity).

14.   I suggest including an integrative figure with the likely mechanism of action of RAC.

Conclusion

15.   It is not concluded, which is the best treatment (5, 10 or 20 μM) of RAC to induce the desired change?

Minor changes

Introduction

1.      Line 46. Please, put the dose of cisplatin that induces interstitial fibrosis in mice.

2.      Methodology: It is not clear how long the cells are left to starve. (line 86)

3.      The passage in which the cell line was used must be added.

4.      How do you switch from low resolution to high resolution in WBs? (from supplemental images).

5.      The manuscript has several old references; please, include more current references.

Discussion

6.      I suggest that Figs 1A from the supplementary material should be incorporated to Fig 4.

7.      Please clarify why Figure 5A only shows data from 10 μM of RAC. What about 5 and 20 μM RAC?

8.      Fig 6. The figure caption must contain the concentration of paclitaxel (10 nM)

Round 2

Reviewer 2 Report

Thank you very much for responding to the suggested comments.